# Quantum Epistemology and Falsification

**DOI:** 10.3390/e24040434

**Published:** 2022-03-22

**Authors:** Giacomo Mauro D’Ariano

**Affiliations:** 1Dipartimento di Fisica dell’Università di Pavia, Via Bassi 6, 27100 Pavia, Italy; dariano@unipv.it; 2Istituto Lombardo Accademia di Scienze e Lettere, 20121 Milano, Italy; 3INFN, Gruppo IV, Sezione di Pavia, 27100 Pavia, Italy

**Keywords:** quantum theory axiomatization, Operational Probabilistic Theories (OPTs), randomness generation, falsifiability

## Abstract

The operational axiomatization of quantum theory in previous works can be regarded as a set of six epistemological rules for falsifying propositions of the theory. In particular, the Purification postulate—the only one that is not shared with classical theory—allows falsification of random-sequences generators, a task classically unfeasible.

## 1. Introduction

Our physical world is ruled by two theories: classical theory (CT) and quantum theory (QT). Compared to CT, QT still looks weird; however, this may be a symptom that we are still missing the hidden logic of the theory. Indeed, we should not forget that among the two theories, QT is the most powerful one, simply because CT is a restriction of QT. In fact, for given system dimension *d*, CT restricts QT’s states to the convex hull of a fixed maximal set of jointly perfectly discriminable pure states (the *d*-simplex), and, correspondingly, transformations are restricted to (sub)Markov linear maps. (For mathematical axiomatizations and main theorems of both theories, QT and CT, see Appendix A). We can thus regard the indeterminism inherent QT as the price to be payed for adding information-processing power.

Deriving QT from information-theoretical principles [1,2,3] reveals how the theory is more powerful than CT. Indeed, the two theories share five postulates, whereas the sixth QT postulate highlights the fundamental task that QT can achieve, whereas CT cannot: *purification*. On the other hand, the sixth CT postulate makes the theory a restriction of QT. Thus, purification synthesizes the additional power of QT compared to CT.

In the following, OPT will be the acronym for Operational Probabilistic Theory. (See Table A1 in Appendix A for acronyms, abbreviations, and symbols). OPTs have been introduced in Refs. [1,2,3], originally inspired by the works of L. Hardy [4] and C. Fuchs [5].

In the convex-OPT language, the five common postulates are:P1*Causality :* The probability of preparation is independent on the choice of observation.P2*Perfect discriminability:* Every state on the boundary of the convex set of states can be perfectly distinguished from some other state.P3*Local discriminability:* It is possible to discriminate any pair of states of composite systems using only local observations.P4*Compressibility:* For all states that are not completely mixed there exists an ideal compression scheme.P5*Atomicity of composition:* The composition of two atomic transformations is atomic.

The sixth postulate, different for each of the two theories, is:P6_*Q*_*Purification:* Every state has a purification. For a fixed purifying system, every two purifications of the same state are connected by a reversible transformation on the purifying system.P6_*C*_*Perfect joint discrimination:* For any system, all pure states can be perfectly discriminated jointly.

Notice that P6${}_{C}$ forces CT to restrict QT’s pure states to a maximal set of perfectly discriminable ones.

## 2. The Purification Principle

Let us recall the statement of the principle.


*For every system A and for every state ρ∈St(A), there exists a system B and a pure state Ψ∈PurSt(AB) such that:*


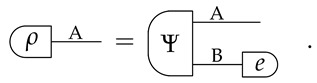

(1)

*If two pure states *Ψ* and $\Psi^{\prime }$ satisfy,*


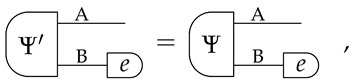


*then there exists a reversible transformation U, acting only on system B, such that:*


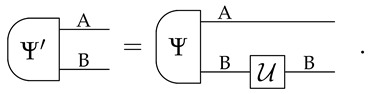

(2)



We call Ψ
*a purification of*
ρ, with B
*purifying system*.

Informally, Equation (Equation 1) guarantees that we can always find a pure state of AB that is compatible with our limited knowledge of A alone. Furthermore, Equation (Equation 2) specifies that all the states of AB that are compatible with our knowledge of A are essentially the same, up to a reversible transformation on B. We call this property *uniqueness of purification*. (Note that the two purifications in Equation (Equation 2) have the same purifying system).

## 3. Epistemological Value of the Postulates

In quantum logic [6], one associates a “proposition” about the system A to an orthogonal projector $P_{S}$ on a subspace $S\subseteq H_{A}$ of the Hilbert space of A. With the map between $P_{S}$ and its support S being a bijection, one can equivalently associate “propositions” to Hilbert subspaces $S\subseteq H_{A}$. (Here, by “support” of an operator we mean the orthogonal complement of its kernel.) One can now enrich the notion of “proposition” by associating it to a quantum state ρ with support Supp ρ=S, the state ρ encoding a reacher information more than just its support S. We conclude that the notion of “state” constitutes a more detailed concept of “proposition” than that of the orthogonal projector. For this reason, in the present context, it is more appropriate to associate the word “proposition” to the notion of “quantum state” instead of the original definition as the orthogonal projector. In other words, a “proposition about A” will be a synonym of the “state of A”. Clearly, the same notion can be extended to CT, since CT is a restriction of QT.

According to the above identification we can now translate QT and CT into the language of “propositions”, and appreciate how, remarkably, all six postulates for QT and CT are of the epistemological nature, i.e., they all are assertions regarding falsifiability of the theory’s “propositions”.

### Epistemological Rules

E1Causality is required for falsification of propositions by observations.E2Perfect discriminability guarantees the existence of falsifiable propositions derived from the theory.E3Local discriminability guarantees that falsifiability of joint propositions can be accomplished “locally”, namely using single system observations and classical communication.E4Compressibility provides the possibility of reducing the dimension of the system, supporting a falsifiable proposition.E5Atomicity of composition guarantees the existence of a class of transformations that do not affect falsifiability of propositions.E6_*Q*_Purification allows for falsifiable random generators.E6_*C*_Perfect discrimination allows any set of determinate propositions to be jointly falsifiable.

Any of the above principles constitutes an epistemological power of the corresponding principle. Statement E6${}_{Q}$, in particular, establishes the possibility of logically falsifying a quantum random generator, namely, there exists a falsification test that establishes if a given quantum random generator is different from a claimed one. Notice that such a falsification cannot be achieved classically for probabilities that are not deterministic, namely, p∉{0,1}, since no succession of outcomes can logically falsify a value of *p* different from 0 and 1. Remarkably, as we will see in this paper, thanks to the purification postulate, within QT we can falsify random generators with any probability distribution. Classically, one falsifies the generator for p=1 whenever the event does not occur, and for p=0 when it does. In such a case, the probability value itself is directly falsified. We stress that in the other cases, QT makes the random generator falsifiable—not the probability value.

Before proving the epistemological rules, we recall the theory of falsification tests introduced in Ref. [7].

## 4. The Falsification Test

We say that an event *F* is a *falsifier* of hypothesis Hyp if *F* cannot happen for Hyp=TRUE. We will call the binary test $\left\{F,F_{?}\right\}$
*a falsification test* for hypothesis Hyp, and denote by $F_{?}$ the *inconclusive event*. Notice that the occurrence of $F_{?}$ generally does not mean that Hyp=TRUE, but that Hyp has not been falsified.

Suppose now that one wants to falsify a proposition about the quantum state ρ∈St(A) of system A. In such case, any effective falsification test can be achieved as a binary *observation test* of the form:(3)$$\left\{F,F_{?}\right\}\subset \mathrm{Eff}\left(A\right),\;F_{?}:=I_{A}-F,\;F>0,F_{?}\geq 0,$$
where by the symbol *F* ($F_{?}$) we denote both event and corresponding positive operator. The strict positivity of *F* is required for effectiveness of the test, F=0 corresponding to the *inconclusive test*, which outputs only the inconclusive outcome. On the other hand, $F_{?}=0$ corresponds to the logical a priori falsification.

Examples of inconclusive tests have been given in Ref. [7], to prove that hypotheses as “purity of an unknown state”, or “unitarity of an unknown transformation” cannot be falsified.

### Falsification of a Quantum State Support

Consider the proposition:(4)$$\;\mathrm{Hyp}:\;\mathrm{Supp}\;\rho =:K⊊H_{A},\;\rho \in \mathrm{St}\left(A\right),\;\;\;\dim H_{A}\geq 2$$
Supp ρ denoting the support of ρ. Then, any operator of the form:(5)$$0<F⩽I_{A},\;\mathrm{Supp}\;F\subseteq K^{⊥}$$
would have zero expectation for a state ρ satisfying Hyp in Equation (Equation 4), which means that occurrence of *F* would falsify Hyp, namely:(6)Tr[ρF]>0 ⇒Hyp=FALSE.
Equation (Equation 5) provides the most general falsification test of Hyp in Equation (Equation 4), the choice $\mathrm{Supp}\;F=K^{⊥}$ corresponding to the most efficient test, namely the one maximizing falsification chance. Notice that the outcome corresponding to I−F does not correspond to a verification of Hyp, since it generally can occur for Supp (I−F)∩K≠∅.

## 5. Proofs of Epistemological Rules

In this section we prove the epistemological rules given in Section 3. We will denote by St(A) the convex set of states of system A and by ∂St(A) its boundary.

E1For any proposition ρ that is falsifiable (i.e., $\mathrm{rank}\rho <d_{A}$), causality protects the falsification target state from being changed by the particular choice of observation.E2Consider two quantum states ρ,ν∈St(A). They are perfectly discriminable iff Supp ρ⊥Supp ν, which implies that $2\leq \mathrm{rank}\rho +\mathrm{rank}\nu \leq d_{A}$ with both ranks at least unit. It follows that $1\leq \mathrm{rank}\rho \leq d_{A}-\min\mathrm{rank}\nu =d_{A}-1$ and the same for rankν, hence both ρ,ν∈∂St(A). We conclude that the two states are falsifiable, with falsifiers kerρ⊆Supp ν and kerν⊆Supp ρ, respectively.E3Here, we will use the *double-ket* notation [8] (for a thorough treatment see [3]). Shortly, once it is chosen the orthonormal factorized canonical basis {|i〉⊗|j〉} for H⊗H, the one-to-one correspondence between vectors in H⊗H and operators on H holds
$$|\Psi \rangle \rangle :=\sum_{ij}\Psi_{ij}|i\rangle ⊗|j\rangle \;⟷\;\Psi =\sum_{ij}\Psi_{ij}\left|i\rangle \langle j\right|\in \mathrm{HS}\left(C^{d}\right),$$
where $\mathrm{HS}(C^{d})$ denotes the Hilbert–Schmidt operators in dimensions *d*. One can then veryify the following identity:
$$\left(A⊗B\right)|C\rangle \rangle =|ACB^{⊺}\rangle \rangle ,$$
with $B^{⊺}$ denoting the *transposed operator* of *B*, the operator that has the transposed matrix w.r.t. the canonical basis. Notice that e.g., ${\left(|\phi \rangle \langle \psi |\right)}^{⊺}=|\psi^{*}\rangle \langle \phi^{*}|$ where $|\psi^{*}\rangle$ is the vector |ψ〉 with complex-conjugated coefficients w.r.t. the canonical basis.Consider now the pure entangled state corresponding to state-vector $|A\rangle \rangle \in H_{A}⊗H_{B}$. The following sequence of identities holds:
(7)$$\left(\langle a\right|⊗\langle b|)|A\rangle \rangle ={(\langle a|⊗I)(I⊗\langle b|A^{T})\sum_{n}|a_{n}\rangle ⊗|a_{n}\rangle }^{*}=\langle b|A^{T}{(|a\rangle }^{*})={\langle b|(A^{†}|a\rangle )}^{*}$$
where $S:=\{|a_{n}\rangle {\}}_{n=1}^{d_{A}}$ is an orthonormal basis for $H_{A}$, with |a〉∈S. Equation (Equation 7) shows that choosing |b〉 orthogonal to $(A^{†}|a\rangle {)}^{*}$ one has a local falsifier of the entangled state |A〉〉〈〈A| given by:
(8)$$P_{A}⊗P_{B}=\left|a\rangle \langle a\right|⊗|b\rangle \langle b|,$$One can see that the generalization to mixtures *R* is straightforward, upon writing the state *R* in the canonical form:
(9)$$R=\sum_{j=1}^{d_{A}}|A_{j}\rangle \rangle \langle \langle A_{j}|,\;\mathrm{Tr}\left(A_{i}^{†}A_{j}\right)=\delta_{ij}p_{j},\;\sum_{j=1}^{d_{A}}p_{j}=1.$$E4Any falsifiable state ρ has dimkerρ≥1, hence it can be isometrically mapped to a state of a system B with $d_{B}\leq d_{A}-1$.E5A transformation A∈Trn(A→B) is called *atomic* if it has only one Krauss term, namely it can be written as $A\rho =A\rho A^{†}$, with $A\in B(H_{A})$ and ||A||≤1. This implies that rank(Aρ)≤rankρ, namely the falsification space has a dimension that is not decreased; hence, the output state Aρ can be falsified. This is not necessarily true for A
*non atomic*, namely with more than one Krauss term, i.e., $A\rho =A_{1}\rho A_{1}^{†}+A_{2}\rho A_{2}^{†}+\ldots$E6_*Q*_See Section 6.E6_*C*_It trivially holds for CT.

## 6. Falsifiable Setup of a Random Generator

A falsifiable setup for a quantum binary random generator can use any quantum system A, e.g., a qubit, in a pure state ρ=|ψ〉〈ψ| along with an orthogonal observation test $\Omega ={\left\{\omega_{i}\right\}}_{i=0,1}$ with $\omega_{i}=\left|i\rangle \langle i\right|$, $\langle i|j\rangle =\delta_{ij}$ (namely a “customary discrete” observable). The following setup,

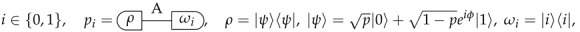
(10)
is a binary random generator with probability $p_{0}=p$.

Notice that the advantage of this choice of setup (compared to, e.g., using a mixed state and/or a non orthogonal observation test) is that it is falsifiable. In fact, the preparation of the state |ψ〉〈ψ| can be falsified efficiently by running the falsification test of the state support, using falsifier F=I−|ψ〉〈ψ|. On the other hand, the observation test Ω can be taken as just the observable providing the physical meaning of the orthonormal basis chosen for the qubit A (e.g., spin-up and spin-down), which is required to physically define the state preparation.

The above setup can be trivially generalized to a *N*-ary random generator (N>2) with probability distribution ${\left\{p_{n}\right\}}_{n\in Z_{N}}$ by using a system A with $\dim(H_{A})=N$, and a pure state with vector with more than two nonvanishing probability amplitudes. Here, F=I−|ψ〉〈ψ| still provides the most efficient falsifier. Notice that for $\dim(H_{A})>2$ it is also possible to falsify mixed states with rank strictly smaller than $\dim(H_{A})$. Notice that the probability of falsification of a mixed state ρ≠|ψ〉〈ψ| is given by p=1−〈ψ|ρ|ψ〉, and vanishes linearly with the overlap between the declared state |ψ〉〈ψ| and the true state ρ.

## 7. Conclusions

CT and QT are more than theories about the world: they constitute extensions of logic. Famously, von Neumann attempted to prove QT to be a kind of logic: we now know that it is an extension of it, instead. We have seen that QT can be regarded as a chapter of epistemology, being a set of rules for accessibility of falsifications. Thus, more than answering the question “what is reality”, QT provides rules for “how we can explore reality”. One can then add axioms to those of QT to get more refined theories, such as Free Quantum Field Theory. The latter can indeed be obtained upon considering a denumerable set of QT systems, and adding the axioms of locality, homogeneity, and isotropy of interactions (see, e.g., the review [9]).

## Data Availability

Not applicable.

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
