# Peer review of "Quantum Epistemology and Falsification"

_entropy, 2022, doi:10.3390/e24040434_

Round 1

Reviewer 1 Report

I juts suggest to extend the introduction, by mentioning the related works of other authors.

Author Response

See the attachment, please.

Reviewer 2 Report

The paper suggests a test capable of , at least in my reading, falsifying the hypothesis of randomness for quantized systems. I on purpose write "at least in my understanding" because I am not extremely sure about this result. Maybe the Author could summarize or explain what he means in more detail.

I also suggest giving very explicit and concrete operational physical examples of such potential falsifications. The ones I see discussed are too abstract; at least for me.

The text is very difficult to read because of the heavy use of abbreviations. I suggest to resolver most of them and substitute most of these abbreviations by their long forms.

Also I suggest not to use a notation like F_? (I suggest F_x instead), or resolve the "double-ket" notation previously used by the Author.

I am also afraid that the Author's bibliography is rather solipsistic and, with the exception of a single reference 6, contains only self-citations. That appears to be rather uncommon; because, one might argue, if the Author merely considers his own papers to be relevant, then this particular paper might only be relevant for a very limited audience---in the extreme form himself and his co-authors.

Author Response

CHANGES SUGGESTED BY REFEREES (Referees’ suggestions in Italics)

(All following changes has been suggested by Referee 2, apart from the last one, that has been also suggested by Referee 1).

—————————————————————————————————————

The paper suggests a test capable of, at least in my reading, falsifying the hypothesis of randomness for quantized systems. I on purpose write "at least in my understanding" because I am not extremely sure about this result. Maybe the Author could summarize or explain what he means in more detail.

The paper does not suggesta test capable of falsifying the hypothesis of randomness for quantized systems”. Due to this misunderstanding, and in view of the excessive emphasis for random generators, I changed the title, as: Quantum epistemology and falsification.

—————————————————————————————————————

I also suggest giving very explicit and concrete operational physical examples of such potential falsifications. The ones I see discussed are too abstract; at least for me.

In the paper, in addition to restating each of the six principles of quantum theory in terms of epistemological rules based on falsifiability, I provided indeed a very simple example in Sect. 6. Falsification of a biased quantum coin. This is indeed a very simple and concrete example, involving just the notions of observable and of quantum state.  

—————————————————————————————————————

The text is very difficult to read because of the heavy use of abbreviations. I suggest to resolver most of them and substitute most of these abbreviations by their long forms.

The only abbreviations in the main text are QT, CT, OPTs (for quantum theory, classical theory, and operational probabilistic theories, respectively). Other abbreviations are used in the appendices, which are not needed for understanding the paper, but are complementary, and report axiomatizations and main theorems for both QT and CT used in the paper. The advantage of self-explaining symbols is that one can synthesize the axiomatizations and related theorems in compact tables. 

—————————————————————————————————————

Also I suggest not to use a notation like F_? (I suggest F_x instead), or resolve the "double-ket" notation previously used by the Author.

The notation F_?  has been used in Ref. 4. In addition, the question mark is a more meaningful symbol to denote an inconclusive result, compared to the letter “x” that has no connection with the context.

—————————————————————————————————————

I am also afraid that the Author's bibliography is rather solipsistic and, with the exception of a single reference 6, contains only self-citations. That appears to be rather uncommon; because, one might argue, if the Author merely considers his own papers to be relevant, then this particular paper might only be relevant for a very limited audience---in the extreme form himself and his co-authors.

In the bibliography I added the two original works by L. Hardy [4] and C. Fuchs [5] that inspired the derivation of Quantum Theory from informational principles in Refs. [1-3].

Moreover, small typos have been corrected, and few presentation improvements have been implemented. 

Round 2

Reviewer 2 Report

Both the Authors and myself may agree to disagree about the relevance of their work to the present foundational debate. But that should not block the Authors from publishing their views on the subject. I therefore recommend publication in its present form.